# Effects of N-Acetylcysteine on the Proliferation, Hormone Secretion Level, and Gene Expression Profiles of Goat Ovarian Granulosa Cells

**DOI:** 10.3390/genes13122306

**Published:** 2022-12-07

**Authors:** Taotao Ji, Xiang Chen, Yan Zhang, Kaibin Fu, Yue Zou, Weiwei Wang, Jiafu Zhao

**Affiliations:** 1Key Laboratory of Animal Genetics, Breeding and Reproduction in the Plateau Mountainous Region, Ministry of Education, Guizhou University, Guiyang 550025, China; 2Key Laboratory of Animal Genetics, Breeding and Reproduction, Guiyang 550025, China; 3College of Animal Science, Guizhou University, Guiyang 550025, China

**Keywords:** N-acetylcysteine, ovarian granulosa cells, RNA-seq, mRNA expression

## Abstract

The purpose of this paper was to investigate the effects of N-acetylcysteine (NAC) on the proliferation, hormone secretion, and mRNA expression profiles of ovarian granulosa cells (GCs) in vitro. A total of 12 ovaries from 6 follicular-stage goats were collected for granulosa cell extraction. The optimum concentration of NAC addition was determined to be 200 μM via the Cell Counting Kit 8 (CCK-8) method. Next, GCs were cultured in a medium supplemented with 200 μM NAC (200 μM NAC group) and 0 μ M NAC (control group) for 48 h. The effects of 200 μM NAC on the proliferation of granulosa cells and hormones were studied by 5-ethynyl-2′-deoxyuridine (EdU) assay and enzyme-linked immunosorbent assay (ELISA). mRNA expression was analyzed by transcriptome sequencing. The results indicate that 200 μM NAC significantly increased cell viability and the proportion of cells in the S phase but promoted hormone secretion to a lesser degree. Overall, 122 differentially expressed genes (DEGs) were identified. A total of 51 upregulated and 71 downregulated genes were included. Gene Ontology (GO) and Kyoto Encyclopedia of Genes and Genomes (KEGG) enrichment analyses indicated that the most DEGs were enriched in terms of cell growth regulation, cell growth, neuroactive ligand-receptor interaction, cytokine-cytokine receptor interaction, the cAMP-signaling pathway, and the Wnt-signaling pathway. Seven genes related to granulosa cell proliferation were screened, *IGFBP4*, *HTRA4*, *SST*, *SSTR1*, *WISP1*, *DAAM2*, and *RSPO2*. The above results provide molecular theoretical support for NAC as a feed additive to improve follicle development and improve reproductive performance in ewes.

## 1. Introduction

Qianbei Ma goat, a unique economic animal in Guizhou Province, China, has many advantages, such as high adaptability, resistance to coarse feeding, stable genetic performance, high disease resistance, and good gregariousness, which is of great significance for maintaining the lives and livelihoods of local people. However, research on Qianbei Ma goats is rare. In the process of Qianbei Ma goat production, follicular hypoplasia is one of the factors leading to nonestrus, abnormal estrus, difficult conception, and the low reproductive rate of ewes, which affect the economic benefits of goat reproduction.

N-acetylcysteine (NAC), which has a small relative molecular mass of 163.19 and readily enters cells, can also be converted into metabolites capable of stimulating glutathione synthesis [1,2]. NAC can restore ovarian function, improve oocyte quality, enhance placental function, and regulate hormone production [3,4]. Mice treated with 100 μM) of NAC for 6 months showed an increase in the litter size of adult mice and delayed oocyte aging in aged mice [5]. Dietary 0.07% NAC supplementation benefited the survival of goat embryos at early gestation and affected apoptosis, angiogenesis, cell growth, and arrest-signaling pathways [6,7,8]. In recent years, the role of NAC in animal reproductive performance at the cellular level has been documented. Low doses of antioxidants can be efficiently applied in cell culturing and maintenance [9], reduce the levels of oxidative stress and apoptosis in mouse GCs, and promote granulosa cell proliferation [10]. Supplementation of oocyte maturation medium with 1.5 mM NAC increased the percentage of viable embryos reaching the blastocyst stage of development [3,11]. NAC (200 μM) may promote goat endometrial stromal cell proliferation and delay cell senescence [12]. However, the mechanisms of NAC effects are not well known.

Granulosa cells (GCs) are an important entry point for assessing follicular development. GCs can convert androgens to estrogens, direct progesterone synthesis, and regulate follicular cyclic derivation. During the reproductive period, the oocyte is surrounded by several layers of GCs that are differentiated into mural and cumulus granulosa cells during the final stages of folliculogenesis inside the follicular microenvironment [13], forming the cumulus–oocyte complex. The cumulus granulosa cells are separated from the oocytes by a zona pellucida, thin cytoplasmic protrusions extend from the granulosa cells through the zona pellucida and contact the oocyte plasma membrane [14], and there is bidirectional communication and nutrient transmission between them [15]. GCs transmit signals that regulate the progression of oocyte development, providing the oocyte with growth factors, such as pyruvate, ribonucleotides, and some amino acids via gap junctions, to promote growth and development. When cumulus granulosa cells in the outer layer of the cumulus–oocyte complex were removed during in vitro culture, oocytes showed growth inhibition [16]. In the development of follicles, with the proliferation and differentiation of granulosa cells, secreted estradiol promotes the development of oocytes and follicles, and the progesterone content gradually increases to promote the maturation and ovulation of follicles [17,18]. The gradual increase in the apoptosis of GCs in developing normal follicles will reduce the support for oocyte growth, including the preparation process of meiotic recovery and progression, as well as the acquisition of fertilization and early development capabilities [19]. Consequently, developing approaches to promote granulosa cell proliferation and hormone secretion would be of fundamental value to promoting follicular development and the reproduction of research animals.

In conclusion, focusing on Qianbei Ma goats, NAC supplementary medium was used to culture GCs for analyzing the molecular mechanisms of the reproductive function of female Qianbei Ma goats by detecting proliferation, hormone secretion, and gene expression. This research will provide an essential theoretical basis for exploring the regulatory mechanism of NAC in the molecular mechanism of Qianbei Ma goats’ reproductive function.

## 2. Materials and Methods

### 2.1. Animal and Sampling

Six 24-month-old ewes (Qianbei Ma goats) were supplied by Fuxing Husbandry Co., Ltd. (Guizhou, China) and slaughtered by Fuxing Animal Husbandry staff in October 2021 (reproductive season). Twelve ovaries of these six goats were collected and sterilized by rinsing once using 75% alcohol, and then sterile phosphate-buffered saline (PBS) was used for washing out the alcohol. The PBS was supplemented with 1% penicillin–streptomycin liquid (10,000 U/mL penicillin and 10 mg/mL streptomycin, Servicebio, Wuhan, China). To ensure the activity of the granulosa cells, the ovaries were soaked with precooled sterile PBS and brought back to the laboratory immediately for the isolation of follicular granulosa cells [20]. Before the implementation of the animal slaughter treatment plan, an application was submitted to the Animal Ethics Committee of Guizhou University (Guiyang, China) and permission was obtained.

### 2.2. Cell Culture and Identification

The ovarian surface was disinfected with 75% alcohol and then washed with PBS 5 times to remove the alcohol thoroughly. The ovaries were then placed in a sterile enzyme-free Petri dish with 20 mL of DMEM/F 12 medium containing 10% fetal bovine serum (FBS) (Gibco, Carlsbad, CA, USA) (complete medium). A 10 mL syringe needle was used to puncture the growing follicle (2.5–5 mm) to make the follicular fluid flow into the dish, and the follicles were washed repeatedly with the culture medium in the dish. Then, the liquid in the dish was collected in a 15 mL sterile enzyme-free centrifuge tube. The collected follicular fluid–medium mixture was centrifuged at 1500 rpm for 8 min. Substances adhering to the bottom of the tube were resuspended in PBS, recentrifuged, digested with 0.25% trypsin solution (Servicebio, Wuhan, China) at a 1:1 ratio for 4 min to separate the cells, and centrifuged again at 1500 rpm for 15 min. The liquid in the tube was discarded, and the cell suspension was prepared by gently pipetting the bottom of the tube with medium and pipetting to the cell flask for culture at 37 °C in a 5% CO_2_ and saturated humidity incubator. After 24 h, non-adherent cells were discarded, and the culture was continued with fresh medium until the cells reached 80% confluence. The cells were passaged by digestion with 0.25% trypsin solution and reseeded [21,22].

Follicle-stimulating hormone receptor (FSHR) was localized specifically to the GCs in growing follicles of various sizes [11]; thus, this cell type can be identified by FSHR immunofluorescence [20,23]. Cells were plated at a density of 1 × 10^4^ cells/well into 24-well plates and grown to a confluent area of 80–90% on the bottom of the well plate. Identification experiments were performed according to the SABC-cy3 immunohistochemistry kit instructions (Boster Biological Technology Co., Ltd., Pleasanton, CA, USA). The primary antibody was rabbit anti-FSH receptor antibody (bs-0895R) (Beijing Biosynthesis Biotechnology Co., Ltd., Beijing, China) [24], and it was diluted 500 times. Immunofluorescence ended, and after washing, nuclear staining was performed using DAPI (Solarbio, Beijing, China) at 25 °C for 10 min. With the aid of DAPI nuclear staining, the type and purity of the cells in 24-well plates could be judged. In this experiment, ovarian granulosa cells were labeled with bright red fluorescence, whereas other cells were not labeled. Images were taken using a Nikon Eclipse Ti fluorescence inverted microscope.

### 2.3. CCK-8 Assay

The viability of the cells was detected using the Cell Counting Kit-8 (CCK-8; Dalian Meilun Biotechnology Co., Ltd., Dalian, China). The cells were seeded in 96-well plates at a density of 5 × 10^3^ cells/well for 24 h at 37 °C and 5% CO_2_ at maximum humidity. The original medium in the plate was replaced with a complete medium containing different concentrations of NAC (Solarbio, Beijing, China) added to each well, setting up 3 replicate wells for each concentration (0, 50, 100, 200, 400, 800, and 1600 μM). After 48 h of continued incubation, the medium containing the drug in the wells was aspirated off and washed using PBS in order not to leave NAC, which would interfere with the detection results. Then, all wells were supplemented with 90 μL of fresh medium, followed by an additional 10 μL of CCK8 reagent per well, and the plates were incubated in the dark at 37 °C for 2.5 h. The OD values at 450 nm were read out by a microplate reader (Bio-Rad, Hercules, CA, USA). According to the results, the optimal concentration of NAC was selected as the treatment protocol for the subsequent experiments. Cell viability* (%)  =  [A (treatment) − A (blank)] / [A (Con) − A (blank)]  ×  100 (A: absorbance at 450 nm) [5].

### 2.4. EdU Assay

Cell proliferation assays were performed using the Meilun EdU Cell Proliferation Kit with Alexa Fluor 555 (Dalian Meilun Biotechnology Co., Ltd., Dalian, China). Cells were counted and seeded into 96-well plates at a density of 5 × 10^3^ cells/well for 24 h and then treated with a medium containing 200 μM NAC or without NAC (adding the same volume of PBS). Then, the cells were incubated with 10 μM EdU at 37 °C for a further 3 h incubation. After fixation with 4% paraformaldehyde for 20 min at room temperature, the cells were permeabilized with 0.3% Triton X-100 for 12 min and washed five times with PBS. Subsequently, the PBS was discarded, a 50 μL click reaction cocktail was added to each well and incubated for 30 min, the reaction liquid was discarded and washed, and the nuclei were stained using Hoechst 33342 (1×) for another 10 min. The nuclear dye was finally blotted off and washed five times using PBS for fluorescence detection. Images were captured using a Nikon Eclipse Ti fluorescence inverted microscope. All of the above steps were performed while protected from light. The proliferation rate: proliferation rate (%) = [number of EdU − positive cells (red)/total number of cells (blue)] × 100%. Three images of different fields were taken per well.

### 2.5. Estradiol and Progesterone Content by ELISA

Estradiol (E2) and progesterone (P4) are secretory components that can be detected in a cell culture medium. Cells were seeded in 96-well plates at a density of 5 × 10^3^ cells/well with 100 μL of complete medium. After the cells had adhered for 12 h, the original culture medium was discarded and not washed by PBS, and 200 μM NAC complete medium and complete medium were directly added to the well plates, which were divided into the 200 μM NAC and control groups, respectively. Six culture wells were used per group. After standing for 2 min, a new medium inside three culture wells in each group was collected to be used as a sample for detection at 0 h, and the remaining wells were incubated for 48 h to collect the culture solution. The collected cell culture was centrifuged at 12,000 rpm for 5 min at 4 °C for hormone detection. At the same time, blank control samples were set up, which were the clean culture media of the two groups. The supernatant was used to detect the content of E2 using a goat E ELISA kit (MM-3352301) (Meimian Co., Ltd., Nanjing, China). The detection range of this kit is 3–150 pmol/L. The determination of P4 content in the culture medium was performed with a goat progesterone ELISA kit (MM-3526601) (Meimian Co., Ltd., Nanjing, China). The assay range of the kit is 35–1500 pmol/L. The coefficient of variation within and between batches detected by these two kits should be less than 10% and 12%, respectively.

### 2.6. Transcriptome Sequencing

#### 2.6.1. Sample Preparation and mRNA Sequencing

The GCs were counted and inoculated into six-well plates at 1 × 10^5^ cells/well. Two plates were inoculated and divided into the 200 μM NAC group and the control group, with six repeated culture wells in both groups. After the cells adhered to the wall, the NAC group was cultured with 200 μM NAC complete medium, and the control group was cultured with complete medium. The cells were cultured for 48 h, the culture medium was discarded, the cells were washed with PBS three times, and the cells were lysed with Trizol (Solarbio, Beijing, China) for RNA extraction. The RNA (200 μM NAC group, *N*= 6; control group, *N*= 6) was submitted to Novogene for sequencing: the PCR products were purified (AMPure XP system), and the library quality was assessed on the Agilent Bioanalyzer 2100 system. An Illumina NovaSeq platform was selected to sequence the library preparations.

#### 2.6.2. Transcriptome Data Analysis

Raw reads were filtered, and the Q30 and GC contents were calculated to obtain clean reads for subsequent analysis [25]. HISAT2 was selected to align paired-end clean reads to the Capra hircus (goat) genome. Then, the expected number of fragments per kilobase of transcript sequence per million base pairs sequenced (FPKM) of each gene was calculated. The Pearson correlation coefficient R^2^ was calculated from the FPKM values of all genes across the samples and plotted as a heat map to visually assess the replicate correlation of the samples between or within groups. Differential expression analysis of the two groups was performed using DESeq2, and |log2 (FoldChange)| > 1&padj < 0.05 was set as the threshold for significantly different expressions. Gene ontology (GO) enrichment analysis of the DEGs and testing of the statistical enrichment of DEGs in KEGG pathways were performed using the clusterProfiler R package, and padj < 0.05 was used as the threshold for significant enrichment.

#### 2.6.3. Quantitative Real-Time PCR

In order to verify the reliability of the sequencing analysis results, the relative abundances of the mRNA of 7 selected transcripts were determined by qRT-PCR with the housekeeping gene β-actin as an endogenous reference. The primers were designed online using NCBI and synthesized by Sangon Biotech (Shanghai, China) Co., Ltd. (Table 1). The total RNA (1 μg) used for sequencing was reverse-transcribed into single-stranded cDNA using a RevertAid First Strand cDNA Synthesis Kit (Thermo Fisher, Waltham, MA, USA). A 10 µL reaction system mix consisted of 3.5 µL RNase-free ddH_2_O, 5 µL 2 × RealStar Green Fast Mixture (GenStar, Beijing, China), 0.5 µL cDNA, 0.5 µL forward, and 0.5 µL reverse primers (10 μM). The reaction procedure was run on ABI QuantStudio3 and QuantStudio5 with the following amplification conditions: 95 °C for 3 min; 40 cycles of 95 °C for 30 s, 57 °C for 30 s, and 72 °C for 30 s. To confirm the specificity of the amplification reaction, a melting curve analysis was performed after the last amplification cycle. The relative mRNA abundances were calculated using the 2^−ΔΔCt^ method [26,27].

### 2.7. Construction and Cell Transfection of Overexpression Recombinant Plasmid

The CDS region sequence of the RSPO2 gene (XM_005689159.3) was cloned into the pEGFP-C1 (kan^+^) vector by Tsingke Biotechnology Co., Ltd. (Beijing, China). The pEGFP-C1-RSPO2 and pEGFP-C1 vectors were transformed into DH5α, and α *Competent Escherichia coli*, propagated in LB liquid medium containing one one-thousandth of kanamycin solution (100 mg/mL) and endotoxin-free plasmids, was extracted using an Endo-Free Plasmid Mini Kit (D6950-01) (OMEGA, Guangzhou, China). GCs were cultured as previously described, seeded into six-well plates at 10^5^ per well, and after adherence, 2.5 ug of the pEGFP-C1-RSPO2 overexpression vector plasmid and pEGFP-C1 empty plasmid were transfected into the cells according to LipofectamineTM 2000 Reagent (Thermo Fisher Scientific, Waltham, MA, USA) instructions. The incubation was continued for 48 h to harvest the cells for RNA extraction.

### 2.8. Expression Abundance Detection of Related Genes under RSPO2 Overexpression

Plasmids were transfected into cells and cultured for 48 h, and the total RNA was extracted using Trizol. qRT-PCR was performed as previously described to detect the expressions of related genes *LGR4*, *CTNNB1*, *PCNA*, *CCND1*, *CYP19A1*, and *HSD17B1*. The primer information of related genes is shown in Table 2.

### 2.9. Statistical Analysis

IBM SPSS Statistics 25 software (IBM Corporation, Armonk, NY, USA) was used for statistical analysis. Graphs were drawn using GraphPad Prism 8. All experimental data are expressed as the means  ±  S.D. of three biological replicates, and the criterion for significance was set at *p*  ≤  0.05 [28]. The CCK-8, EdU, and RT-PCR results were analyzed for significance using one-way ANOVA or Student’s *t*-test.

## 3. Results

### 3.1. GC Isolation and Characterization

GCs were successfully obtained using a previously described collection and isolation method. The expression of FSHR in GCs is specific; thus, this cell type can be identified by FSHR immunofluorescence [20,23]. The initial seeding density of the cells was 10^4^/well in 24-well plates, and they were cultured for 48 h until cell confluence reached 80% for testing. Three fields were selected in each well; the fluorescence was observed and photographed. Figure 1 shows that FSHR was positively expressed in the cytomembrane (Figure 1A), all nuclei were blue (Figure 1B), and the first two graphs coincided completely (Figure 1C), which indicates that the purity of the GCs isolated from goats was high.

### 3.2. The Effect of NAC on Cell Proliferation

The results of the CCK-8 assay show that different concentrations of NAC had different effects on the goat granulosa cells in vitro. The optimum concentration of NAC was 200 μM. Low concentrations of NAC had a positive effect on cell viability; when the concentration was 800 μM or 1600 μM, it did not promote cell vitality or even inhibit cell vitality. The cell viability of the 200 μM group was significantly higher than those of the 50 μM, 100 μM, and control groups (*p* < 0.05). The cell viability of the 400 μM group was significantly higher than that of the 50 μM and control groups. Treatment with 200 μM NAC had the highest effect on proliferation (Figure 2A). Embedding EdU into S-phase cells revealed that the percentage of cell proliferation in the 200 μM group was significantly higher than that in the control group (Figure 2B).

### 3.3. Estradiol and Progesterone Levels

The ELISA results show that the contents of E2 in the control group and 200 μM NAC group increased significantly or extremely significantly 48 h later, and the contents of progesterone also increased very significantly. At 48 h, the contents of estradiol and progesterone in the 200 μM NAC group were higher than those in the control group, but there was no significant difference between the two groups (Figure 2C,D).

### 3.4. Transcriptomic Analysis of Cell Samples of the 200 μM NAC and Control Groups

#### 3.4.1. Total RNA Quality Detection and Data Quality Control Analysis

A total of 12 transcriptome libraries were constructed from six samples of the 200 μM NAC group (experimental group (EG): GCS_NAC1, GCS_NAC2, GCS_NAC3, GCS_NAC4, GCS_NAC5, and GCS_NAC6) and six samples from the control group (CG: GCS1, GCS2, GCS3, GCS4, GCS5, and GCS6). The total RNA integrity values of the samples were >8.80. Via quality control, an average of 42.30 million clean reads were generated (39.50 million to 47.22 million for the 200 μM NAC group, and 40.19 million to 45.61 million for the control group). The mean GC content of the 12 samples was 52.23. The Q30 values were greater than 93.32% and less than 95.04%, meeting the requirement of 90% for Q30 (Table 3). On average, 96.14% of the clean reads were mapped to the Capra hircus (goat) genome (taxid: 9925), and 87.78% to 92.12% were uniquely mapped to the Capra hircus (goat) genome (taxid: 9925). All of the above results indicate that the quality of the sequencing results was good to underpin all analyses regarding the expression profiles obtained for the sequencing data (Table 3).

#### 3.4.2. Gene Expression

To clarify the effect of NAC on granulosa cell gene expression, differentially expressed genes between the 200 μM NAC group and the control group were analyzed. In total, 12,155 expressed genes (FPKM > 1) were identified in the two groups. Among them, 208 genes were only expressed in the control group, 171 genes were only expressed in the 200 μM NAC group, and 11,776 genes were expressed in both groups (Figure 3A). The Pearson correlation coefficient (R2) between the samples met the requirement of being greater than 0.92, which reflects the reproducibility of the experiment and guarantees the reliability of the subsequent differential gene analyses (Figure 3B). A total of 122 differentially expressed genes (DEGs) were screened in the 200 μM NAC group. There were 51 genes whose expression was significantly upregulated, and 71 genes whose expression was significantly downregulated. Cluster analysis of the DEGs showed that the samples of the 200 μM NAC group and control group had good intragroup repeatability (Figure 3C,D). Red represents a higher level of gene expression, and green represents a lower level of gene expression.

#### 3.4.3. GO Functional and KEGG Pathway Enrichment Analysis of DEGs

The GO analysis identified the DEGs and divided them into three parts: biological processes (BPs), cellular components (CCs), and molecular functions (MFs). There were 26 terms that were significantly enriched (Figure 4A); 7 belonged to biological processes, 1 to cellular components, and 18 to molecular functions, indicating that the DEGs were mainly enriched in molecular functions. In biological processes, upregulated genes were mainly significantly enriched in the regulation of cell growth, cell growth, the regulation of growth, growth, and the regulation of cellular component organization. In cellular components, upregulated genes were mainly significantly enriched in the extracellular region. In molecular functions, upregulated genes were mainly significantly enriched in insulin-like growth factor binding, growth factor binding, and enzyme inhibitor activity (Figure 4B). However, downregulated genes were only significantly enriched in molecular functions, such as phosphoprotein phosphatase activity, phosphatase activity, chemokine activity, and chemokine receptor binding (Figure 4C). Based on the gene annotations in the terms, the highlighted candidate genes were *IGFBP4*, *WISP*, and *SST*.

The KEGG pathway analysis showed that the 122 DEGs of the two groups were involved in 33 pathways. The KEGG enrichment results are insignificant if padj < 0.05 is used as the threshold for significant enrichment. Therefore, according to the padj value from small to large, the first 20 KEGG pathways were selected to draw a scatter plot for display (Figure 4D). To investigate the effect of 200 μM NAC on cell proliferation and hormone secretion, several important pathways related to cell growth and reproduction were screened. They were neuroactive ligand–receptor interaction, the cAMP-signaling pathway, cytokine-cytokine receptor interaction, the cell cycle, and the Wnt-signaling pathway (Figure 4D), which are closely related to cell growth and reproduction. Combining GO and KEGG analyses, seven DEGs related to granulosa cell proliferation and hormone secretion were identified, including *IGFBP4*, *HTRA4*, *SST*, *SSTR1*, *WISP1*, *DAAM2*, and *RSPO2*. Among these genes, five genes (*IGFBP4*, *HTRA4*, *WISP1*, *DAAM2*, and *RSPO2*) were upregulated, and two genes (*SST* and *SSTR1*) were downregulated.

### 3.5. RNA-seq Data Validation

The relative abundance of the three upregulated genes, *MFAP4*, *IGFBP4*, and *DUSP27*, and the four downregulated genes, *MYO7A*, *CCNB1*, *RGS5*, and *CSPG4*, which had high fold change according to the sequencing results, were detected by qRT-PCR. As expected, the qRT-PCR results are basically consistent with the RNA-seq results (Figure 5), with the *MFAP4*, *IGFBP4*, and *DUSP27* mRNA levels being increased in the 200 μM NAC group by qRT-PCR. Similarly, the levels of *MYO7A*, *CCNB1*, *RGS5*, and *CSPG4* were reduced in the 200 μM NAC group. These results indicate that the sequencing data are reliable.

### 3.6. Regulation of Overexpression of RSPO2 on GCs

To evaluate the regulation of the *RSPO2* gene on GCs, qRT-PCR was used to detect the related gene expression abundance of cells after transfection with the *RSPO2* overexpression vector plasmid. According to the qRT-PCR results, the overexpression of *RSPO2* in cells significantly promoted the expression of the *RSPO2* receptor gene *LGR4*, the LGR4 target gene *CTNNB1*, the cell-proliferation-related genes *PCNA* and *CCND1*, and the estradiol synthesis related gene *CYP19A1*, but had no significant effect on *HSD17B1* gene expression (Figure 6).

## 4. Discussion

In this study, when GCs were cultured in different concentrations of NAC medium (50 μM to 1600 μM) for 48 h, with the increase in NAC concentration, the cell viability first increased and then decreased. This might be attributed to the nature of NAC itself. NAC is a weak organic acid, and when the concentration of NAC in the solution is too high, its pH decreases, damaging the cell membrane [29]. E2 and P4 are secreted by GCs and are involved in the regulation of granulosa cell proliferation and differentiation as well as the formation and development of follicles [30,31]. In the present study, when the cells were in the logarithmic growth phase, the secretion of E2 and P4 was greatly increased in both groups. This result is consistent with that obtained when using 100 μM NAC to act on ovine ovarian granulosa cells in vitro [32] and consistent with the law of hormone secretion [33]. However, overall, the secretagogue effect of 200 μM NAC on the hormone was not significantly different from that of the control group. The amounts of E2 and P4 were much lower than the physiological concentrations in estrus or ovulation [34].

The changes in granulosa cell proliferation and hormone secretion only represent a phenotype induced by NAC. It is of fundamental significance to explore the mRNA expression changes behind this phenotype through transcriptome sequencing technology. Integrating the GO and KEGG results, the genes involved in regulating cell proliferation were mainly *IGFBP4*, *HTRA4*, *SST*, *SSTR1*, *WISP1*, *DAAM2*, and *RSPO2*. According to previous studies, these genes play an important role in the growth, hormone secretion, luteinization, and follicular development of granulosa cells.

In antral follicles, insulin-like growth factors (IGFs) are known to stimulate granulosa cell proliferation and steroidogenesis in most mammals. The bioavailability of IGFs is regulated by a family of IGF-binding proteins (IGFBPs) expressed inside antral follicles [35]. IGFBP-4 mRNA expression varies between species. *IGFBP4* is persistently expressed in healthy follicles and luteinized GCs, does not inhibit follicle growth, and is involved in the steroidogenesis or luteinization of granulosa cells. Immunohistochemistry of ovarian sections from female rhesus monkeys revealed IGFBP4 gene expression in the GCs of a mature dominant follicle [36]. In porcine follicle studies, *IGFBP4* was similar in growing follicles, such as 2, 4, and 6 mm follicles, but increased in both GCs and theca interna cells in 8 mm follicles, which were mature follicles [37]. *HTRA4*, at the N-terminus, has a similar domain structure to that of IGFBPs; it is recognized as an IGF binding site and plays key roles in cell growth and cell proliferation [38]. In in vivo animal experiments, feeding 0.07% NAC to Nubian goats upregulated the mRNA expression of insulin-like growth factor and its binding protein in ovarian tissue [39]. It is thus speculated that NAC upregulated the mRNA expression of these two genes, affecting the insulin-like growth factor (IGF) system and thereby regulating granulosa cell proliferation. Studies have shown that SSTs and SSTRs negatively regulated follicle development via GCs. *SST* is present in follicular fluid and is associated with progesterone secretion [40]. Inhibiting the expression of *SST* and *SSTR* would reduce their negative binding to adenylate cyclase, thus increasing the concentration of cyclic adenosine monophosphate (cAMP) and promoting the proliferation of GCs [41,42]. Moreover, the coculture system of bovine oocytes and granulosa cells supplemented with SST resulted in a significant reduction in oocyte maturation [43]. In the present study, NAC downregulated the expression of *SST* and *SSTR* in granulosa cells under the effect of promoting granulosa cell proliferation, which may lend support to the above findings about the regulation of *SST* or *SSTR*.

The transcriptome data show that the Wnt pathway was especially enriched with three genes, *WISP1*, *DAAM2*, and *RSPO2*, all of which showed upregulated expression under the effect of 200 μM NAC. They are key genes regulating granulosa cell proliferation and hormone secretion. Numerous studies have shown that the Wnt-signaling pathway has an important role in female reproduction. *WISP1* is the downstream key target gene of the Wnt/β-catenin pathway [44], which is involved in cell adhesion, survival, proliferation, differentiation, and migration and is frequently reported to be associated with tumors. *DAAM2* regulates crucial aspects of cell fate determination, cell migration, cell polarity, neural patterning, and organogenesis during embryonic development and can interact with *DVL* to regulate the Wnt-signaling pathway [45]. It plays a role in regulating embryonic development, but little is known about its role in the female ovary. The R-spondin family are secreted activators of the WNT/β-catenin (*CTNNB1*)-signaling pathway. *RSPO2* is described as a key factor in folliculogenesis regulation working in a paracrine manner and maintaining granulosa cell proliferation [46,47,48,49]. In mice, the deletion of the *RSPO2* gene does not damage the growth of oocytes but blocks the cell cycle progression of neighboring GCs, leading to follicular growth stagnation. In a study of goats, *RSPO2* was found to have a potential relationship with ovarian differentiation [48]. *CYP19A1* is a key signaling enzyme in the synthesis of estradiol. Zhou et al. [50] indicated that *RSPO2* promotes E2 secretion in porcine GCs by increasing the expression levels of *CYP19A1* and *HSD17B1*. Moreover, *RSPO2* knockdown promoted the apoptosis of GCs, blocked follicle development, and delayed the onset of puberty by reducing the expression levels of *LGR4* and *CTNNB1,* which are Wnt-signaling-pathway-related genes in vivo. These are similar to the results of the present study: overexpressed *RSPO2* promoted the expression of the receptor *LGR4*, causing the expression of the downstream target gene *CTNNB1* within the Wnt-signaling pathway. The upregulation of *PCNA* and *CCND1* expression regulated the cell cycle to promote cell proliferation, thus driving *CYP19A1* expression to favor estradiol synthesis. The specific functions of other candidate genes regulating granule cells need to be further investigated. These research data lay a foundation for studying the regulatory effect of NAC on animal reproductive performance.

## 5. Conclusions

The above results show that 200 μM NAC upregulated *IGFBP4*, *HTRA4*, *WISP1*, *DAAM2*, and *RSPO2* genes and downregulated *SST* and *SSTR1* genes in GCs, which mainly act on neuroactive ligand–receptor interaction, cytokine-cytokine receptor interaction, the cAMP-signaling pathway, and the Wnt-signaling pathway to regulate cell proliferation and hormone secretion. However, the specific mechanisms of these actions require more in-depth verification. This study provides a reference for the further investigation of the effects of NAC on animal reproduction at the cellular level.

## Figures and Tables

**Figure 1 genes-13-02306-f001:**
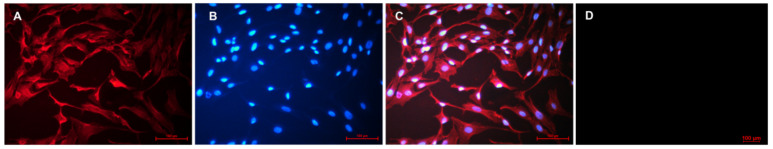
Representative immunofluorescence images of fixed GCs. (**A**) GCs were stained with cy3 to visualize FSHR (shown in red). (**B**) DNA was stained with DAPI (shown in blue). (**C**) Merged image of (**A**) and (**B**). (**D**) Blank control without FSHR antibody staining. The fluorescence in the cells in the same field was observed under a fluorescence inverted microscope under ultraviolet and green excitation light and camera. *N* = 3.

**Figure 2 genes-13-02306-f002:**
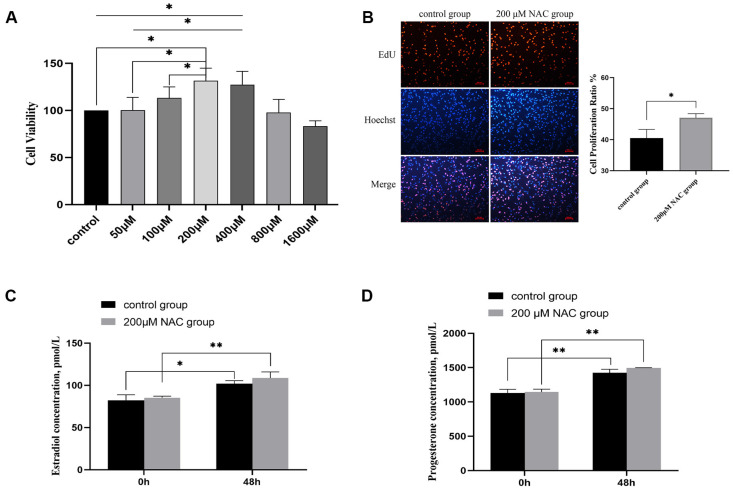
Regulation of NAC on the growth of granulosa cells. (**A**) Effect of different concentrations of NAC on cell proliferation. (**B**) Cell proliferation of 200 μM NAC and control groups was detected by EdU assay at 48 h. EdU: The thymidine analog EdU in the kit replaces thymidine and infiltrates the newly synthesized DNA in the S-phase of the cell cycle. After the Click reaction, the newly synthesized DNA was labeled with red fluorescence. Hoechst: Hoechst 33342, Hoechst coloration presented light blue, round morphous, and formed fairly dark particles inside normal cell nuclei; Merge: Superimposing and merging the red fluorescence and blue fluorescence images in the same field of view. Histogram of cell proliferation ratio: The proportion of cell proliferation in the 200 μM group was significantly higher than that in the control group. ***** Indicates a significant difference between the two groups at the 0.05 level. (**C**) Estradiol levels. Estrogen contents in the culture medium of the 200 μM NAC group and control group were detected at 0 h and 48 h, respectively, *n* = 3. (**D**) Progesterone levels. Progesterone contents in the culture medium of the 200 μM NAC group and control group were detected at 0 h and 48 h, respectively, *n* = 3. * Indicates a significant difference between the two groups at the 0.05 level. ** Indicates a significant difference between the two groups at the 0.01 level.

**Figure 3 genes-13-02306-f003:**
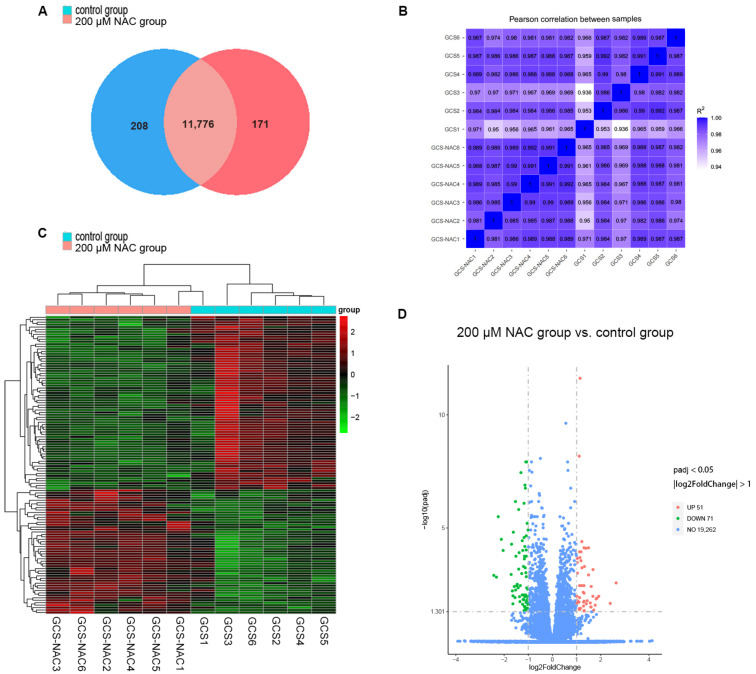
Analysis of gene expression. (**A**) Venn diagram of gene expression. (**B**) Pearson correlation between samples. (**C**) Heat map of the DEGs cluster analysis. Columns represent samples, and different rows represent genes. (**D**) Volcano plot of DEGs (|log2(FoldChange)| ≥ 1 and padj < 0.05). Up, upregulated DEGs; Down, downregulated DEGs. GCS-NAC1, GCS-NAC2, GCS-NAC3, GCS-NAC4, GCS-NAC5, and GCS-NAC6 are representative of the 200 μM NAC group. GCS1, GCS2, GCS3, GCS4, GCS5, and GCS6 are representative of the control group.

**Figure 4 genes-13-02306-f004:**
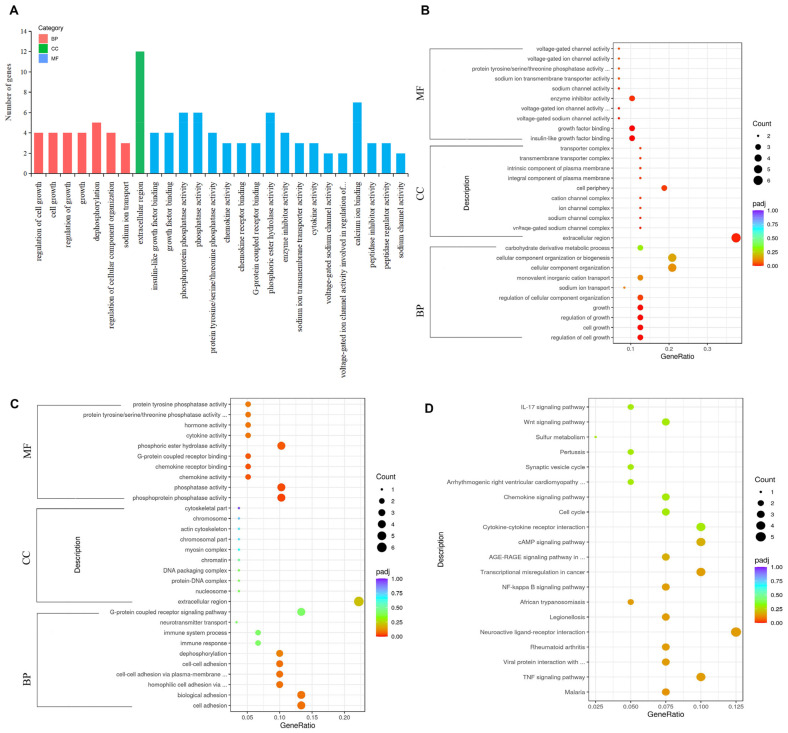
GO and KEGG enrichment analyses of DEGs of GCs between the 200 μM NAC group and control group. (**A**) Statistics of GO enrichment. Histogram plots were generated with all differential significance terms. The abscissa is the GO term, and the ordinate is the number of genes enriched by the GO term. (**B**) Dot plot of upregulated gene-enriched GO terms. (**C**) Dot plot of downregulated gene-enriched GO terms. (**D**) Dot plot of top 20 KEGG pathways. The size of the dots represents the number of genes annotated onto GO terms and the KEGG pathways, and the colors from red to purple represent the significant size of enrichment.

**Figure 5 genes-13-02306-f005:**
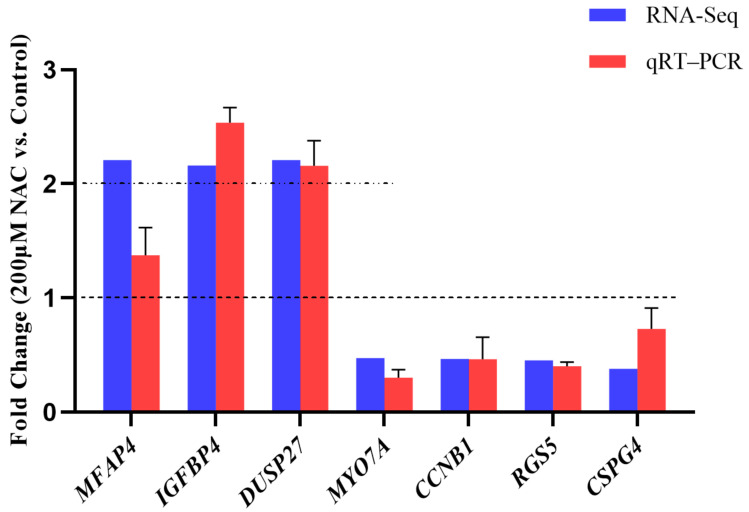
Relative abundance of 7 DEGs. RNA-Seq: Gene expression abundance obtained from transcriptome sequencing data, expressed as fold change. When the value corresponding to a gene is greater than 2, the gene is significantly expressed in the 200 μM NAC group. If it is greater than 0 but less than 1, it is expressed significantly in the control group. qRT-PCR: Gene expression abundance by qRT-PCR. 2^−ΔΔCt^ was taken as the fold change. When the value corresponding to a gene is greater than 1, the gene is significantly expressed in the 200 μM NAC group. *N* = 3.

**Figure 6 genes-13-02306-f006:**
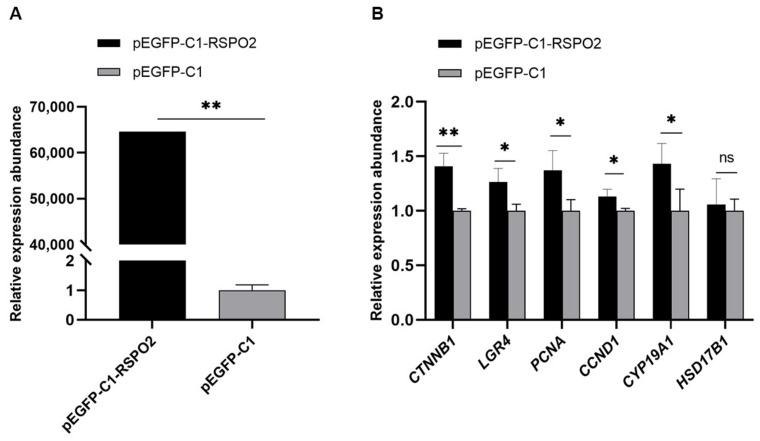
Regulation of *RSPO2* gene on GCs. (**A**) *RSPO2* overexpression efficiency validation. (**B**) Expression abundance of related genes. * Shows that there is a significant difference at the 0.05 level among groups, ** Indicates a significant difference at the 0.01 level. The genes to be tested were normalized using housekeeping genes, the relative expression abundance was expressed as 2^−ΔΔCt^, and the results are shown as means ± SD (*N* = 3).

**Table 1 genes-13-02306-t001:** The primers for qRT-PCR.

Gene Name	Login ID	Oligonucleotide Primers (5′-3′)	Product Size (bp)
*MFAP4*	XM_018064747.1	F: GGGGCCCAGCACATAACTC R: ACGACTTCTCCAGAGCATCTCC	187
*IGFBP4*	NM_001285678.1	F: GGCCAAAATTCGAGACCGGA R: TAAAGGTCTTCGTGGGTGCG	162
*DUSP27*	XM_018046440.1	F: TGGAGGACCTGTACAACCGT R: TTGTTCACAGCCACACTCTTCT	178
*MYO7A*	XM_018059307.1	F: CTGTCAGGGATGGAGTCGTG R: AGTGCACTTGTGGTAGCCTC	158
*CCNB1*	XM_005694606.3	F: GCTGTTGGCCTGTGGTTGAT R: GCATTAATCTTCGTGTTCCTGGT	130
*RGS5*	XM_005677115.3	F: CCAAGACCCAGAAACCCTCG R: TTTGCCGTCTCAGCCATCTT	199
*CSPG4*	XM_018066268.1	F: TATGAGCACGAGATGCCCTC R: TCGGAGACCCAGAGACCTTT	179
*β-actin*	XM_018039831.1	F: CCGTGACATCAAGGAGAAGC R: CCGTGTTGGCGTAAAGGT	266

*MFAP4*: microfibrillar-associated protein 4, *IGFBP*4: insulin-like growth factor binding protein 4, *DUSP27*: dual-specificity phosphatase 27 (putative), *MYO7A*: myosin VIIA, *CCNB1*: cyclin B1, *RGS5*: regulator of G-protein signaling 5, *CSPG4*: chondroitin sulfate proteoglycan 4.

**Table 2 genes-13-02306-t002:** The primers for qRT-PCR of related genes.

Gene Name	Login ID	Oligonucleotide Primers (5′-3′)	Product Size (bp)
*RSPO2*	XM_005689159.3	F: ACCATGTCCGACCATTGCTGAATC R: GTGTTGCTCCTGGGCTCTTTCTATC	142
*CTNNB1*	XM_018066894.1	F: CGGCAATCAAGAAAGCAAGCTCATC R: CACAGACAGCACCTTCAGCACTC	130
*LGR4*	XM_018059305.1	F: GCAGTCCGCCCACTCTGATTATG R: CTCTTCCGCAAGCCTGAACCTG	83
*PCNA*	XM_005688167.3	F: GAAGAAAGTGCTGGAGGC R: TCGGAGCGAAGGGTTA	129
*CCND1*	XM_018043271.1	F: GCTGCGAGATGGAAAC R: AAGTAGGACACCGAGGG	115
*CYP19A1*	NM_001285747.1	F: CCCAAGGCATTACAATGT R: TAAGGGTTTCCTCTCCAC	266
*HSD17B1*	XM_018065125.1	F: GGAGCATAGGCGGCTTGAT R: TGGCGACAGTAGCGGTAGAA	241

*RSPO2*: R-spondin 2, *CTNNB1*: catenin β 1, *LGR47*: leucine-rich repeat-containing G protein-coupled receptor 4, *PCNA*: proliferating cell nuclear antigen, *CCND1*: cyclin D1, *CYP19A1*: cytochrome P450 family 19 subfamily A member 1, *HSD17B1*: hydroxysteroid 17-β dehydrogenase 1.

**Table 3 genes-13-02306-t003:** Quality control and sequencing data statistics.

Samples	RIN	Clean Reads	GC Content/%	Reads Quality ≥ Q30/%	Mapped Reads/%	Unique Mapped Reads/%
GCS-NAC1	8.80	42,795,496	51.96	94.14	95.80	90.45
GCS-NAC2	9.50	47,220,802	52.37	95.04	97.38	92.12
GCS-NAC3	9.40	40,030,012	52.94	94.54	96.48	91.51
GCS-NAC4	9.00	41,594,026	52.45	94.62	96.50	91.26
GCS-NAC5	8.90	39,502,108	52.76	94.73	96.54	91.62
GCS-NAC6	9.90	45,110,510	52.21	93.91	96.72	91.50
GCS1	9.10	45,609,488	49.60	93.32	93.17	87.78
GCS2	10.00	42,688,270	52.84	94.72	96.33	91.15
GCS3	10.00	41,268,492	52.99	94.27	96.16	90.82
GCS4	10.00	40,585,386	52.38	94.65	96.16	90.85
GCS5	9.40	41,043,756	52.52	94.81	96.40	91.38
GCS6	8.90	40,193,144	51.78	94.78	95.98	90.56

Six samples of 200 μM NAC group: GCS-NAC1, GCS-NAC2, GCS-NAC3, GCS-NAC4, GCS-NAC5, and GCS-NAC6; six samples from the control group: GCS1, GCS2, GCS3, GCS4, GCS5, and GCS6.

## Data Availability

The raw data from sequencing are available at NCBI under BioProject ID PRJNA867504.

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
