# Peer review of "Effects of N-Acetylcysteine on the Proliferation, Hormone Secretion Level, and Gene Expression Profiles of Goat Ovarian Granulosa Cells"

_genes, 2022, doi:10.3390/genes13122306_

Round 1

Reviewer 1 Report

The manuscript has been evaluated carefully. In this study, the authors aimed to investigate the effect of N-acetylcysteine on the proliferation, hormone secretion, and mRNA expression profiles of caprine ovarian granulosa cells in vitro. Please note my specific  comments below:

L45: Please be more precise. Which females are the authors referring to?

L60+62: If the authors introduce the abbreviation first in L60, there is no need to re-introduce it in L62. Please fix.

Introduction: Please add more background information on the physiology of cumulus cells, too. There is missing information on key processes related to the interplay between the oocyte and surrounding cells.

L81-83: Please explain why only six goats have been slaughtered although ten are mentioned earlier.

L229-241: Please indicate the number of replicates and the cell count number.

L406-430: This section should be moved to the introduction.

General comments on the study design: Please explain how the GCs have been distributed to the treatment (different concentrations of NAC)  and control group.

If the authors did use only six goats, 12 ovaries were used to collect GCs. Did the author perform a crucial pre-selection on the cycle stage of the goats? How did the authors select the ovaries for GCs collection based on the number of visible follicles, CLs? That should be all clarified to analyze appropriately the outcome. Please add also an exemplary picture of the caprine ovary.

Reviewer 2 Report

In this paper the authors investigate the impact of NAC treatment on hormone secretion in goat ovarian cells. This is relevant question as goat reproductivity is one of the factors affecting goat farmers, whilst NAC's role in reproduction has been previously shown in mice. However several sections of the paper are poorly described and could benefit of few additional controls (see comments below). In addition, abstract does not clearly indicate what the objective of the study was nor what the state-of-the-art is, and as such does not convey why the study has been done and what the relevance of the results are.

Comment 1: Section 3.1 Expression of FSHR - the description of the figure is poor and there are typos in the figure legend. Line 106-109 (in materials and methods) should be included in section describing Figure 1 to make clear, what the reader is looking at. The fact that blue and red colour "coincide" is not an indication that the cell population is pure. Ideally, staining for a protein that should not be expressed in this cells could be done.

Comment 2: Section 3.2 It is not clear what the lowercase letters in the figure legend indicate.

Comment 3:  It is not clear how authors chose various time points they look at, e.g. why hormones are checked at 6h-48h, while transcriptomics are done only at 48h. It should either be clarified, or these measurements should be done at matching time points, or there should be a biological justification as to why hormone secretion is measured at earlier time points. There also should be a clear indication in the figure legend, and not just in methods section, what time points are presented in the figures, e.g. figure 1A.

Comment 4: Gene ontology description is very poor and should be re-written. For example: there is nothing special in GO terms being divided into three terms, and instead the focus should be on which descriptors of the given ontology are interesting, significant/relevant. 

Comment 5: It is unclear if the estradiol / progesterone production levels were normalized to the differences in the proliferation and viability. If yes, it should be clarified, if not cell count at different time points should be used as a reference, as cell number WILL affect the levels of hormones that are secreted.

Comment 6: Authors produced a list of the most relevant DE genes in response to NAC. The levels of these genes should be confirmed by WB, if possible. In addition, the role of these genes in NAC-induced changes in proliferation/viability/hormone secretion should be tested, by either overexpression or siRNA.

Round 2

Reviewer 1 Report

No further comments.

Author Response

Thank you very much for your valuable suggestions and guidance to improve our manuscript.

Reviewer 2 Report

Figure 1 legend - this is still not up to standards. One cannot say "red fluorescence was expressed in cytoplasm", instead e.g. "Representative immunofluorescence images of fixed GC cells. Cells were stained with ... to visualise X protein (shown in red), DNA was stained with DAPI (blue)" might be used.

Figure 2 b - I still have no idea what the lowercase letters mean. The significance between different parameters can be displayed easily by connecting the relevant bars with lines.

In the abstract the objective of the paper is set as: "to clarify the potential mechanisms of NAC", this is not a scientific justification - at the stage of reading an abstract it is not clear what NAC is and mechanism of what action of NAC are being studied.

Authors did not comment on my suggestion regarding cell number when assaying hormone production - has this been accounted for?
